# Single-molecule tracking of the transcription cycle by sub-second RNA detection

Zhengjian Zhang[1]*[†], Andrey Revyakin[1]*[†‡], Jonathan B Grimm[1], Luke D Lavis[1], Robert Tjian[1,2,3]

[1]Janelia Farm Research Campus, Howard Hughes Medical Institute, Ashburn, United States; [2]Li Ka Shing Center for Biomedical and Health Sciences, University of California, Berkeley, Berkeley, United States; [3]Department of Molecular and Cell Biology, University of California, Berkeley, Berkeley, United States

**Abstract** Transcription is an inherently stochastic, noisy, and multi-step process, in which fluctuations at every step can cause variations in RNA synthesis, and affect physiology and differentiation decisions in otherwise identical cells. However, it has been an experimental challenge to directly link the stochastic events at the promoter to transcript production. Here we established a fast fluorescence in situ hybridization (fastFISH) method that takes advantage of intrinsically unstructured nucleic acid sequences to achieve exceptionally fast rates of specific hybridization ($\sim 10e7$ $M^{-1}s^{-1}$), and allows deterministic detection of single nascent transcripts. Using a prototypical RNA polymerase, we demonstrated the use of fastFISH to measure the kinetic rates of promoter escape, elongation, and termination in one assay at the single-molecule level, at sub-second temporal resolution. The principles of fastFISH design can be used to study stochasticity in gene regulation, to select targets for gene silencing, and to design nucleic acid nanostructures.

*For correspondence: zhangzh@ janelia.hhmi.org (ZZ); revyakina@ janelia.hhmi.org (AR)

[†]These authors contributed equally to this work

[‡]**Present address:** University of Leicester, Leicester, United Kingdom

**Reviewing editor**: James T Kadonaga, University of California, San Diego, United States

## Introduction

Transcription is the first, and frequently the most regulated step in the flow of genetic information from DNA to protein. Transcription is a dynamic, multi-step process in which RNA polymerase (RNAP) (i) binds to the promoter to form the closed complex ($RP_c$); (ii) melts the promoter to form the open complex ($RP_o$); (iii) performs several abortive cycles of synthesis and release of short 2–12 nt RNA products as the initial transcribing complex ($RP_{itc}$); (iv) escapes the promoter; (v) undergoes some promoter-proximal pausing; (vi) forms an elongating complex ($RD_e$) whose processivity can either be interrupted by more pauses or stimulated by trailing RNAPs and elongation factors; and, finally (vii) terminates at the end of the transcription unit (reviewed in *DeHaseth et al., 1998*; *Murakami and Darst, 2003*; *Cheung and Cramer, 2012*). Due to the low copy number of genes in a cell (usually one in prokaryotes and two in eukaryotes), molecular fluctuations at any of the above steps may cause large cell-to-cell variability in the amount of the final RNA transcript produced in populations of otherwise genetically identical cells grown under identical conditions, and thus can affect gene expression, and cell physiology (*Ogawa, 1993*; *Raj and Van Oudenaarden, 2009*; *Yamanaka, 2009*; *Gupta et al., 2011*; *Lionnet and Singer, 2012*). Therefore, understanding how molecular fluctuations at different steps of the transcription cycle alter transcriptional outcomes is required to dissect the mechanism of gene regulation.

Single-molecule techniques are ideally suited to directly monitor molecular fluctuations in multi-step reactions in real-time, without averaging out their inherent stochasticity (*Weiss, 1999*), and have provided important insights into the dynamics of transcription, unattainable by conventional ensemble

**eLife digest** The body produces proteins by transcribing DNA (genes) to make messenger RNA, which is then translated to make a protein. Transcription begins when an enzyme called RNA polymerase binds to the DNA and catalyzes the process by which genetic information from the double helix is copied to a complementary RNA transcript, which subsequently becomes the messenger RNA.

Because a living cell usually contains only one or a few copies (alleles) of a given gene, molecular fluctuations play a crucial role in cellular transcription. Therefore, studying transcription kinetics at the level of single molecules may provide critical insights into how cells deal with—or even take advantage of—molecular fluctuations. A number of different single-molecule techniques can be used to follow transcription, but these techniques are often relatively slow compared to transcription in living cells, or they suffer from other problems such as only being able to study one step in the transcription process.

Now, Zhang, Revyakin et al. have systematically devised a technique called 'fastFISH' that is fast enough to track the production of single RNA molecules directly and instantaneously. FastFISH builds on an existing technique called FISH—short for fluorescence in situ hybridization—in which fluorescent molecules are attached to single strands of DNA or RNA. These single strands pair with specific regions of complementary DNA or RNA molecules, and they can be visualized with a fluorescence microscope. However, conventional FISH is a 'snap-shot' technique that is not suitable for making real-time observations under physiological conditions.

FastFISH relies on single strands of fluorescently labeled DNA and RNA that bind to complementary strands of DNA or RNA extremely quickly, even under physiological conditions, because they contain only three of the four 'regular' nucleotides that make up DNA or RNA. As a proof of principle, Zhang, Revyakin et al. used fastFISH to study the kinetics of transcription by the bacteriophage T7 RNA polymerase and were able to measure multiple stages of the transcription cycle in a single-molecule experimental setup.

By allowing each stage of transcription to be tracked in real-time at the level of single-molecules, fastFISH will permit a more in-depth analysis of the factors that regulate how genes are expressed as proteins in our cells. Moreover, the ability to design single-strand probes that bind rapidly to DNA and RNA targets could have many additional applications, including new strategies for more efficient gene silencing.

methods (*Bai et al., 2006*). For instance, single-molecule assays based on optical nanomanipulation have revealed pausing and backtracking of $RD_e$ (*Neuman et al., 2003*; *Shaevitz et al., 2003*; *Galburt et al., 2007*) and measured the force and the torque generated by $RD_e$ (*Wang et al., 1998*; *Ma et al., 2013*). Methods based on magnetic nanomanipulation and single-molecule fluorescence resonance energy transfer have probed conformational changes in DNA and RNAP within $RP_o$, and $RP_{itc}$ (*Kapanidis et al., 2006*; *Revyakin et al., 2006*; *Tang et al., 2009*; *Chakraborty et al., 2012*; *Robb et al., 2013*). Finally, single-molecule localization studies have probed the dynamics of the initial promoter search by RNAP (*Wang et al., 2012*; *Friedman et al., 2013*). However, most current single-molecule methods focus on only one step of transcription, and are not well suited to relate protein-DNA complex assembly and dynamics during multiple stages of the transcription cycle to RNA production. In addition, most methods do not measure RNA production directly, but rather infer it from changes in DNA conformation or movement of RNAP along the DNA. Most recently, single-molecule tracking of key protein-DNA interactions coupled with detection of the RNA production has been demonstrated in the bacterial (*Friedman and Gelles, 2012*) and human transcription systems (*Revyakin et al., 2012*). The former study achieved time resolution for RNA detection on the order of ~10 s, and thus provided a dynamic, quantitative view of the full transcription cycle of bacterial RNAP. However, time scales of many events in transcription are on the order of ~1 s, particularly under physiological conditions (for instance, the residence time of transcription factors on DNA (reviewed in *Hager et al., 2009*), the rate of initiation and promoter clearance by RNAPs (*Revyakin et al., 2006*; *Tang et al., 2009*), and the expected time delay between cooperatively elongating RNAPs molecules (*Epshtein and Nudler, 2003*)). Thus, a real-time method for nascent transcript detection at ~1 s time scales would significantly enhance our

ability to dissect the dynamics of the transcription process, and to correlate stochastic fluctuations at different steps with transcriptional outcomes.

Currently, the most sensitive and specific methods for detecting RNA transcripts continue to rely on complementary nucleic acid hybridization. However, oligonucleotide probes typically hybridize several orders of magnitude slower than diffusion under physiological conditions (effective rate constant $k_{on}$ less than $10^5$ M$^{-1}$ s$^{-1}$, [*Chan et al., 1995* and references therein]) and, as a result, do not permit real-time nascent RNA detection with common single-molecule setups, such as total internal reflection fluorescence (TIRF) microscopy. Here we present a fast fluorescence in situ hybridization method (fastFISH) to detect synthesis of nascent RNA transcripts at sub-second time resolution, at the single-molecule level. The method takes advantage of our finding that single-stranded nucleic acid probes with sequences comprised of just three of the four bases (A, U and C for RNA probes, and A, T, and G for the complementary DNA targets) are intrinsically unstructured and, as a result, hybridize at exceptionally fast rates (~$10^7$ M$^{-1}$s$^{-1}$) without compromising sequence specificity. As a proof of concept, we applied fastFISH to probe the production of nascent RNA transcripts by the speedy bacteriophage T7 RNA Polymerase (T7 RNAP) in vitro. Furthermore, by using fastFISH in combination with fluorescently labeled RNAP, we dissected the full T7 RNAP transcription cycle (promoter binding, promoter escape, elongation, and termination). FastFISH can be used to study transcription by multi-subunit prokaryotic and eukaryotic RNA polymerases at the level of stochastic molecular interactions. The rules for generating fastFISH probe-target pairs can also be used for gene silencing, gene profiling and bottom-up assembly of nucleic acid nanostructures (*Rothemund, 2006*).

## Results

### Design and characterization of fastFISH hybridization probes

To achieve real-time nascent RNA detection, we set out to find a general rule for designing RNA-probe pairs that hybridize at the fastest possible rates. Hybridization of two single-stranded nucleic acid fragments requires unfolding of the fragments into unstructured coils, which then anneal to form an intermolecular base paired helix (*Lima et al., 1992* and references therein). In support of this notion, nucleic acid probes with less stable secondary structures show faster hybridization rates to complementary nucleic acid targets (*Lima et al., 1992*; *Kushon et al., 2001*; *Wang and Drlica, 2004*; *Gao et al., 2006*; *Yilmaz et al., 2006*). Likewise, decreasing the length of hairpins in conventional molecular beacons increases their hybridization rates (*Tsourkas et al., 2003*). Thus, we reasoned that intrinsically unstructured nucleic acid oligonucleotide pairs of optimal lengths should have the fastest hybridization kinetics under physiological conditions without compromising specificity.

To systematically examine the propensity of RNA sequences to form base-paired secondary structures, we used the nucleic acid structure prediction tool Mfold (*Zuker, 2003*) to calculate the free energy of self-folding, $\Delta G_{37°C}$, of randomly selected short RNA sequences (N = 338,417). We chose RNA sequence with length of 19 nt and GC content between 0.4 and 0.6 as representative of a typical oligonucleotide primer. We found that $\Delta G_{37°C}$ values were mostly negative and widely distributed ($-1.7 \pm 1.7$ kCal mol$^{-1}$), indicating that an average random 19-mer RNA sequence is mostly structured (*Figure 1A*). Next, we closely examined a subset of 19-mer RNA sequences that had positive $\Delta G_{37°C}$ (>+2 kCal mol$^{-1}$, N = 2,768, <1% of the total pool of random sequences), and found that these mostly unstructured sequences were composed predominantly of only three bases A, U, and C (~84% had only 2 G residues or less, significantly less than the 4.75 G residues expected on average, *Figure 1—figure supplement 1*). The bias towards the lower G-content in the unstructured sequences was not surprising, because guanine is the most potent in base-paring interactions: it forms a triple hydrogen bond with cytosine and a wobble pair with uracil. We then calculated $\Delta G_{37°C}$ for randomly picked 19-mers composed of only A, U, and C (N = 99,777, GC content between 0.4 and 0.6, *Figure 1A*), and found that these AUC-sequences had mostly positive $\Delta G_{37°C}$, with a much narrower distribution (+2.5 ± 0.6 kCal mol$^{-1}$, *Figure 1A*). Analysis of other three-base-derived RNA sequences (AUG, CAG, CUG) indicated that AUC-sequences were unique in their largely positive $\Delta G_{37°C}$ (*Figure 1A*). Calculation of $\Delta G_{37°C}$ for random DNA 19-mers (that could be used as complementary probes for RNA targets) indicated that ATG-based and ATC-based DNA 19-mers were also significantly less structured than their four-base counterparts (*Figure 1A*). Therefore, we hypothesized that the use of the AUC alphabet for RNA targets, and ATG alphabet for DNA probes should allow the fastest annealing reactions under physiological temperature of 37°C.

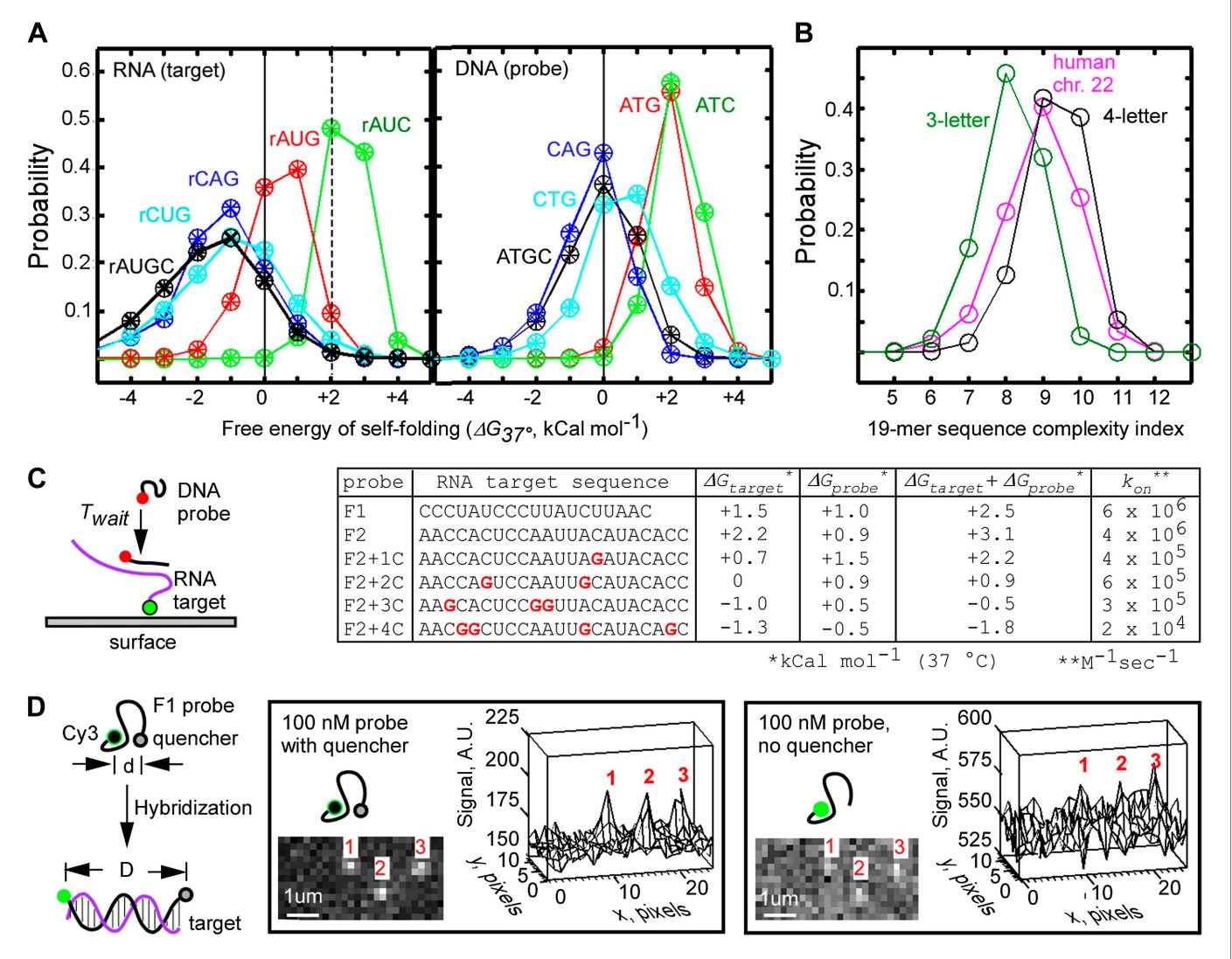

**Figure 1**. Design of fastFISH probe-target pairs. (**A**) Probability distribution of Mfold-calculated free energies of self-folding of randomly selected, single-stranded 19-mer RNA (left) and DNA (right) oligonucleotides, composed of three or four bases. Results of analysis of three independent sets are shown as '+', 'x', and '○'. About 100,000 three-letter sequences, and about 300,000 four-letter sequences were analyzed in each set. (**B**) Lempel–Ziv complexity analysis of three-letter 19-mer oligonucleotides (one set of ~100,000 AUC sequences), four-letter 19-mer oligonucleotides (one set of ~300,000 AUGC sequences), and all tiling 19-mers from the exome of the human chromosome 22. (**C**) Single-molecule measurements of hybridization rates of fastFISH probe-target pairs, and the effect of G-residues in the targets on the rates. Left: schematic of experiment. Cy3-labeled 90-base RNA oligonucleotides containing a single target sequence were immobilized on a glass surface through a biotin moiety at the 3′ end. Atto633-labeled DNA probes were injected into the imaging flow cell, and their hybridization was detected using TIRF/CoSMoS to obtain the probe arrival time $T_{wait}$. Right: table of RNA target sequences, Mfold-calculated free energies of self-folding of RNA targets ($\Delta G_{target}$), DNA probes ($\Delta G_{probe}$), combined energies of targets and probes, and on-rates calculated based on probe $T_{wait}$ and concentrations. (**D**) Self-quenching approach to reduce fluorescence background from unbound DNA probes in TIRF imaging of probe-target hybridization. Left: schematic of experiment. A quencher (e.g., Iowa Black FQ) is placed on one end of a DNA probe, and a fluorophore (e.g., Cy3) is placed on the opposite end of the DNA probe. The short persistence length of single-stranded DNA ($l_o$ ~0.8 nm, **Smith et al., 1996**; **Dessinges et al., 2002**) enables quenching of Cy3. Upon hybridization to the target, the distance between Cy3 and Iowa Black FQ increases due to the larger $l_o$ ~50 nm of the DNA-RNA duplex, leading to an increase of Cy3 fluorescence. Middle: representative TIRF image and a corresponding three-dimensional plot of target-hybridized F1 probes acquired in the presence of unbound, self-quenched F1 probe at 100 nM. Right: same set of molecules imaged in the presence of unbound, unquenched F1 probe at 100 nM.

The following figure supplements are available for figure 1:

**Figure supplement 1**. Sample sequences of the unstructured 19-mer RNA oligos.

*Figure 1. Continued on next page*

*Figure 1. Continued*

**Figure supplement 2**. Single-molecule measurements of fastFISH probe-target hybridization rates using TIRF/CoSMoS.

**Figure supplement 3**. Reduction of fluorescence background of free fastFISH probes by self-quenching: additional representative images.

To ensure that the use of the three-base alphabet did not compromise the specificity of probe-target hybridization, we calculated the textual complexity of randomly picked three-letter 19-mers using the algorithm of Lempel and Ziv (*Ziv and Lempel, 1976*; *Kaspar and Schuster, 1987*; *Orlov and Potapov, 2004*). This algorithm, commonly used in lossless data compression programs, calculates the minimal number of operations required to reconstruct a sequence of symbols by copying and inserting segments of an existing sub-sequence. Thus, nucleic acid sequences that contain repetitive elements would have a lower LZ complexity (and would be less specific in hybridization) (*Wright and Church, 2002*). We found that three-letter 19-mer sequences had complexity indices of $8.1 \pm 0.8$ (N = 99,777), while four-letter 19-mers had complexity indices of $9.3 \pm 0.8$ (N = 338,417, *Figure 1B*). As a comparative reference, we calculated complexity indices for tiling 19-mers in all exons of human chromosome 22 to be $8.9 \pm 1.0$ (*Figure 1B*). Importantly, a significant fraction of three-letter 19-mers (~31%) had complexity indices of 9 and higher, which matched or exceeded the average complexity of human exons. These calculations indicate that a significant fraction of the random probe sequences composed of only three bases nevertheless retain sequence complexity and specificity that match their physiological, four-base derived counterparts.

We applied the AUC rule and the complexity filter to generate two candidate probe-target pairs (referred to as F1 and F2). The calculated $\Delta G_{37°C}$ for the F1 and F2 pairs were +1.5 and +2.2 kCal mol$^{-1}$, respectively, for the AUC-based RNA targets, and +1.0 and +0.9 kcal mol$^{-1}$ for their ATG-based complementary DNA probes (*Figure 1C*), suggesting that the pairs are mostly unstructured and likely to be fast-hybridizing. Indeed, the F1 and F2 probes annealed to their surface-immobilized RNA targets at exceptionally fast rates—$6 \times 10^6$ M$^{-1}$s$^{-1}$ and $4 \times 10^6$ M$^{-1}$s$^{-1}$, respectively, as measured by TIRF-based Colocalization Single-Molecule Spectroscopy (CoSMoS, *Friedman et al., 2006*) (*Figure 1C*, *Figure 1—figure supplement 2*). These on-rates were at least 100-fold faster than typical four-base nucleic acid probes of similar lengths reported in literature (*Lima et al., 1992*; *Kushon et al., 2001*; *Wang and Drlica, 2004*; *Friedman et al., 2006*; *Gao et al., 2006*; *Yilmaz et al., 2006*). Consistent with the AUC rule, introducing back one or more G residues into the F2 RNA target sequence decreased the rate of its hybridization to the complementary DNA probe (10-fold reduction for one G residue and 300-fold reduction for four G residues), which correlated with the progressively lower free energies of self-folding (*Figure 1C*, *Figure 1—figure supplement 2D*). We conclude that the target sequences designed using the AUC rule, combined with the complexity filter, achieved our goal of generating superior hybridization kinetics for fast transcript detection.

Although the F1 and F2 probes were exceptionally fast, detection of RNA on the sub-second time scales would require the use of fluorescent probe concentrations above 100 nM (*Figure 1—figure supplement 2D*). Such concentrations are generally incompatible with common single-molecule detection setups such as TIRF microscopy, due to high fluorescence background from the freely diffusing, unbound probe molecules. We overcame this problem by attaching a quencher to the single-stranded probe at the end opposite of the fluorophore (*Marras et al., 2002*). This self-quenching strategy enabled the use of free probes at concentrations up to 1 µM (*Figure 1D*, *Figure 1—figure supplement 3*). We believe that, in the absence of the secondary structure, the self-quenching effect was likely mediated by random polymer motion and/or contact quenching (*Johansson et al., 2002*; *Marras et al., 2002*). We refer to our method for fast nucleic acid detection using unstructured, sequence specific, self-quenched fluorescent probes as 'fastFISH'.

## Real-time single-molecule detection of transcription with fastFISH

To demonstrate that fastFISH can detect nascent transcripts in real time at single-molecule resolution, we used the bacteriophage T7 RNAP. This single-subunit RNAP is an excellent test case for fastFISH: at physiological temperature (37°C) it initiates transcripts at an effective rate of at least 1 s$^{-1}$ (*Martin and Coleman, 1987*), and elongates transcripts at ~240 nt s$^{-1}$ (*Golomb and Chamberlin, 1974*; *Bonner et al., 1994*). As a model template, we used a fluorescently labeled linear DNA fragment

containing the consensus promoter for T7 RNAP (*Milligan et al., 1987*) and the F1 target sequence (positioned between +28 and +46 from the transcription start site, +1), and ending at +295. We surface-immobilized the Cy3-labeled DNA via a biotin moiety attached to the upstream end of the template (position −75) (*Figure 2A*). In this configuration, RNAP molecules were expected to initiate transcription at the promoter, elongate the nascent RNA in the direction away from the surface, and run off the free, untethered template end at +295, together with their nascent RNA products.

We supplied unlabeled RNAP, NTPs and the fluorescent, self-quenched F1 probe to the imaging flow-cell, and monitored the interactions of the F1 probe with the DNA loci by TIRF/CoSMoS (*Friedman et al., 2006*; *Revyakin et al., 2012*). We reasoned that the F1 target sequence in the nascent RNA would become available for hybridization once the active site of elongating RNAP ($RD_e$) reaches position +60 (taking into account the ~14 bases of nascent RNA protected by RNAP [*Huang and Sousa, 2000*]). The $RD_e$ complex would then remain on the DNA until it runs off at +295. At the expected average RNAP elongation rate of 240 nt s$^{-1}$, the F1 probe would have a brief ~1 s window of opportunity to 'catch' the nascent transcript at the DNA molecule of origin. Probe hybridization to the RNA product would then be observed as a fluorescent spot co-localizing with the DNA locus.

*Figure 2B* shows the co-localization analysis of probe-DNA interactions in a typical single-molecule fastFISH experiment. This analysis counts all DNA molecules that co-localized with a probe 'spot' for more than five cumulative frames (at 0.4 s/frame) during the incubation (~15 min), and plots the $\Delta x, \Delta y$ displacements between the probe and DNA molecules as a two-dimensional histogram (*Revyakin et al., 2012*). Specific probe-DNA interactions were observed, as indicated by a prominent peak at position (0, 0) of the co-localization histogram. In a typical experiment, 10–30% of DNA templates in a

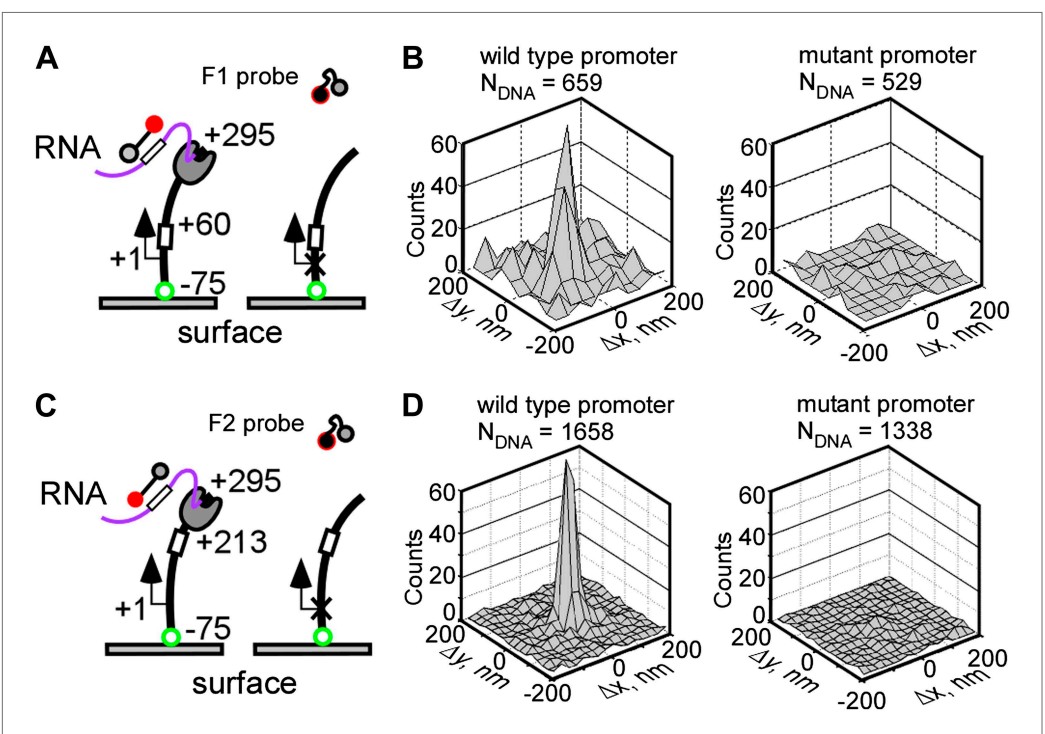

**Figure 2**. Real-time single-molecule detection of transcription by T7 RNAP using fastFISH. (**A**) Schematic of experiment. DNA templates containing a single consensus promoter for T7 RNAP (+1), or a single null mutant promoter (cross), were immobilized on a surface, with the promoter directing transcription towards the free end (+295). The templates contained the F1 fastFISH target sequence downstream from the promoter (from +28 to +46) which was expected to become available for hybridization in the nascent RNA after the RNAP active site reaches position +60. (**B**) Co-localization analysis of F1 probe-DNA interactions in a representative experiment. Left: wild type promoter. Right: mutant promoter. (**C**) and (**D**) Same as panels (**A**) and (**B**), but for the F2 target and probe. The templates contained the F2 fastFISH target sequence (from +181 to +199) which was expected to become available for hybridization in the nascent RNA after the RNAP active site reaches position +213.

field of view co-localized with a probe. This apparent incomplete template utilization was not due to inefficient probe hybridization (see *Figure 4* and 'Discussion'), but, instead, was likely limited by the accessibility of the DNA template due to surface interference. No probe-DNA interactions were observed in a control experiment with a DNA template containing a null mutant promoter (*Raskin et al., 1993*), indicating that the co-localization was due to promoter-specific transcription. We also performed real-time single-molecule transcription experiments using the other unstructured probe, F2, and obtained nearly identical DNA-probe co-localization results (*Figure 2C–D*). Taken together, these findings suggest that fastFISH can detect production of nascent transcripts in real-time.

## Single-molecule dynamics of T7 RNAP-DNA interactions

To measure the efficiency of real-time detection of nascent transcripts by fastFISH, we needed a reference to define the start and end of each productive single-molecule transcription event. We reasoned that monitoring the interactions between RNAP molecules and the DNA templates during promoter binding and run-off can serve this purpose. Therefore, we fluorescently labeled RNAP with Cy5 using HaloTag (*Figure 3—figure supplement 1*), supplied Cy5-RNAP and NTPs to an imaging flow cell containing immobilized DNA templates, and monitored Cy5-RNAP interactions with the DNA molecules (*Figure 3*). We observed transient RNAP binding events that lasted between 0.08 and 8 s, and were promoter-specific (*Figure 3B–D*). The events were comprised of two distinct populations: short-lived events whose dwell times fit well to a single exponential distribution (mean dwell time $T_0$ ~0.14 s); and a long-lived, bell-shaped population skewed towards long-lived events (peak dwell time $T_1$ ~1.7 s). The durations of observed interactions were not limited by the lifetime of the Cy5 label before photobleaching (*Figure 3—figure supplement 2*). We interpret the short-lived, stochastic interactions to be promoter-specific but non-productive $RP_c$ and $RP_o$, because no interactions were observed with templates containing a null promoter sequence (at frame rates of 12.5 Hz, *Figure 3B*) and similar short-lived interactions were observed in the absence of NTPs (single exponential $T_0$ ~0.3 s, *Figure 3D*).

We interpret the long-lived, bell-shaped population of events to be full, productive transcription cycles, because (i) the long-lived events were not observed in the absence of NTPs (*Figure 3D*); (ii) the peak dwell time of the long-lived events, $T_1$, increased nearly linearly with the length of the transcribed DNA segment ($T_1$ ~1.7 s, ~2.7 s, and ~3.7 s for DNA segments with $l$ = 295 bp, 633 bp, and 910 bp, respectively, *Figure 3E–G*); and (iii) the RNAP fluorescence signal, on average, decayed towards the end of long-lived events (*Figure 3C,H*), consistent with RNAP initiating transcription at the promoter located close to the surface (75 bp, or ~25 nm), and then elongating towards the free, untethered DNA end, down the gradient of the TIRF evanescent field.

We conclude that the dwell time distributions of RNAP-DNA interactions define productive transcription events which can be used as a reference to determine the efficiency of nascent RNA detection by fastFISH, and to dissect the full transcription cycle by RNAP. We note that the slope of the plot of long-lived RNAP dwell times vs DNA length, (~3.2 ± 0.6) × $10^{-3}$ s nt$^{-1}$ (*Figure 3G*), gives an estimate of RNAP elongation rate of ~300 nt s$^{-1}$. The intercept of the plot with the time axis ($l$ = 0) gives an estimate of the net time that RNAP spends on the DNA template without elongating, $T_{stationary}$ = 0.7 ± 0.3 s, which includes the net duration of promoter opening and abortive cycling ($RP_c$, $RP_o$ and $RP_{its}$), and, possibly, the time RNAP idles at the free DNA end before run-off.

## Single-molecule dissection of the T7 RNAP transcription cycle with fastFISH

To determine the efficiency of nascent RNA detection by fastFISH, and to demonstrate the use of fastFISH in dissecting the kinetics of the full transcription cycle, we simultaneously monitored RNAP-DNA interactions and the production of transcripts in real-time, using Cy5-labeled RNAP together with the Cy3-labeled probe F1 (*Figure 4A*). In this two-color experiment, we expected to observe the following sequence of events: (i) initial appearance of an RNAP spot at the DNA locus, corresponding to promoter binding ($RP_c$); (ii) continued occupancy of the DNA by RNAP, corresponding to promoter opening ($RP_o$), abortive cycling ($RP_{itc}$), elongation to +60 ($RD_e$) and the time required for probe hybridization; (iii) appearance of an F1 probe spot; (iv) co-occupancy of the DNA by RNAP and probe corresponding to elongation ($RD_e$); and (v) simultaneous disappearance of the RNAP and probe spots corresponding to run-off at +295. As shown in a representative video montage, precisely this sequence of events was observed (*Figure 4B,C*). Typically, the probe arrived at the DNA within 1 s after arrival of RNAP.

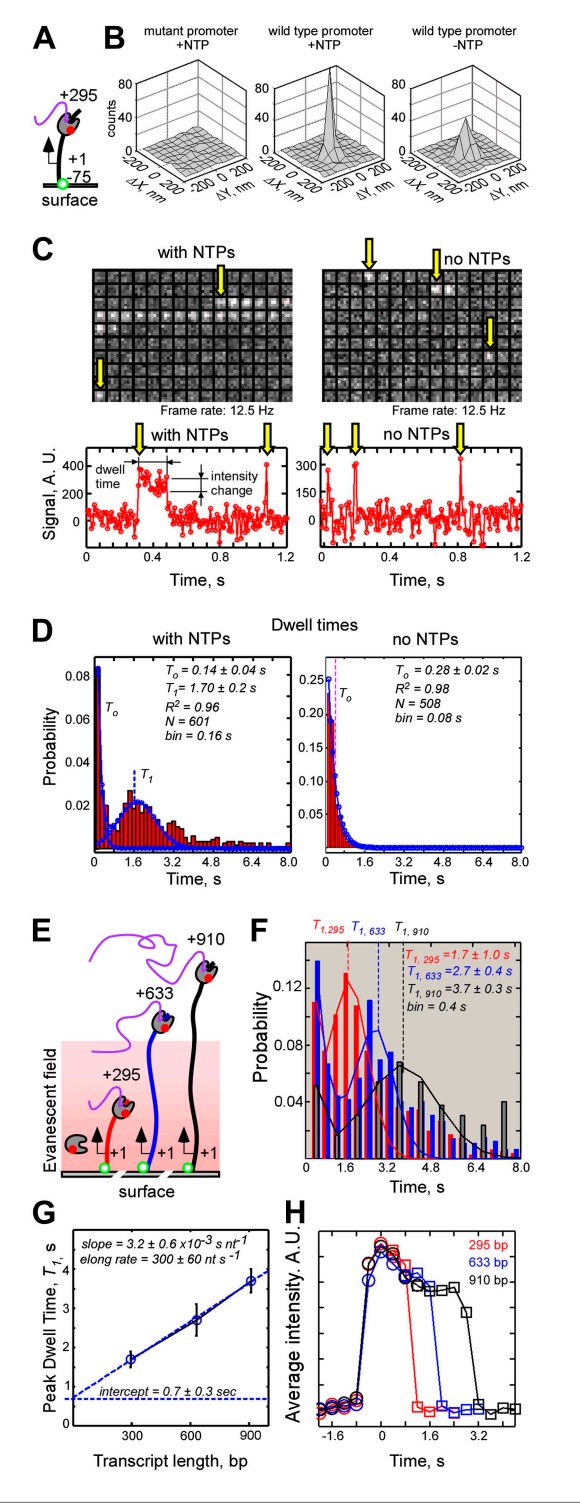

**Figure 3**. Single-molecule dynamics of T7 RNAP-DNA interactions. (**A**) Schematic of experiment. (**B**) Co-localization analysis of RNAP-DNA interactions. Left: null promoter DNA template in the presence of NTPs. Center: wild type promoter DNA template in the presence of NTPs. Right: wild type promoter DNA template in the absence of NTPs. (**C**) Representative data. Top: video montages of RNAP

*Figure 3. Continued on next page*

Three lines of evidence suggested that the temporal co-occurrence of the F1 probe and RNAP corresponded to specific capture of nascent RNA in $RD_e$. First, no binding of F1 after RNAP binding was observed in the absence of NTPs (data not shown), or in the presence of only ATP and GTP, which limited initial transcription to positions from +1 to +6 (***Figure 4—figure supplement 1***), indicating that the probe bound specifically to the nascent RNA, and not to RNAP or to the DNA template. Second, no F1 probe binding was observed in the absence of the target sequence in the transcribed segment of the template, indicating that hybridization was specific for the F1 target (***Figure 4—figure supplement 2***). Third, and most importantly, when we moved the F1 target sequence from position +28 to position +181, we found that binding of the probe was delayed by an additional ~0.5 s (***Figure 4—figure supplement 3***), consistent with the extra time $RD_e$ needs to transcribe the extra 153 bp DNA segment at 300 nt s⁻¹ (***Figure 3G***).

To estimate the efficiency of nascent RNA detection by fastFISH, we post-synchronized (***Blanchard et al., 2004***) all RNAP-DNA and probe-DNA interactions by RNAP run-off at the end of productive transcription events (defined as RNAP interactions that lasted ≥0.8 s), and super-imposed them to generate heat maps of all RNAP-DNA and probe-DNA interactions (***Figure 4E***). We found that ~81% of $RD_e$s contained an RNA probe immediately before run-off (***Figure 4F***). The residual ~19% of undetected $RD_e$s were mostly likely due to the presence of dark fluorophores in the chemically synthesized probes (***Revyakin et al., 2012***), and the inherent stochasticity of probe hybridization. The heat map analysis of all RNAP-DNA and probe-DNA interactions also showed that, in the majority of cases (~72%), the probe and RNAP dissociated from the DNA simultaneously upon run-off ($\Delta T_{off}$ ~0). In the remaining ~28% of cases, the RNA probe persisted on the DNA for ≥0.4 s after RNAP run-off (***Figure 4F***); the nature of these events remains to be determined.

To extract the kinetic information on early stages of transcription, we post-synchronized all RNAP-DNA and probe-DNA interactions by RNAP binding during

*Figure 3. Continued*

interactions with template containing consensus promoter for a 5 × 5 pixel region of interest centered at a single, photobleached DNA molecule (1 pixel = 200 nm, imaged at 12.5 Hz). Bottom: fluorescence time traces corresponding to the montages shown on top. Baseline of zero intensity indicates no binding. Left: experiment carried out in the presence of NTPs. Right: experiment carried out in the absence of NTPs. Yellow arrows indicate the first frames of RNAP binding events. (**D**) Dwell time probability distributions of RNAP-DNA binding events. Left: experiment carried out in the presence of NTP. Fitting to a sum of single exponential and Gaussian functions is shown in blue. Right: experiment carried out in the absence of NTPs. Fitting to a single exponential function is shown in blue. (**E**) Dependence of the peak dwell time of RNAP-DNA interactions on the length of the transcribed DNA segment: schematic of experiment. DNA templates containing transcribed segments spanning from +1 to +295 (red), +633 (blue), or +910 (black) were separately immobilized, and interactions of labeled RNAP were recorded at 2.5 Hz. (**F**) Dwell time probability distributions of RNAP-DNA interactions for the three DNA templates shown in (**E**). N = 749 for the +1…+295 template (red), N = 716 for the +1 … +633 template (blue), and N = 213 for the +1 … 910 DNA template (black). The peak dwell times were calculated by fitting the distributions to a sum of single exponential and Gaussian functions. (**G**) Plot of the peak dwell time of RNAP-DNA interactions vs the length of the transcribed DNA segment. (**H**) Decay of intensity of RNAP fluorescence signal during productive RNAP-DNA interactions as an indicator of elongation by RNAP. All RNAP-DNA interactions having dwell times longer than 0.8 s (experiments in **F**) were post-synchronized by RNAP binding (t = 0, circles) and by RNAP run-off (squares), and weight-averaged plots of RNAP binding and run-off were plotted for DNA templates having transcribed segments of different lengths (red – 295 bp; blue – 633 bp; black – 910 bp). Time offsets between RNAP binding and run-off were set at peak dwell lifetimes, $T_1$, for the respective DNA templates measured in (**F**).

The following figure supplements are available for figure 3:

**Figure supplement 1**. Labeling and biochemical characterization of T7 RNAP activity.

**Figure supplement 2**. Measurements of photobleaching times of fluorophores used in this study.

productive transcription (RNAP-DNA interactions that lasted ≥0.8 s) (**Figure 4D**). Analysis of the resulting heat-map of probe and RNAP binding gave the average time delay between the RNAP and probe arrival, $\Delta T_{on}$ of 0.7 ± 0.2 s. We interpret $\Delta T_{on}$ as the net duration of promoter opening ($T_{open}$), abortive cycling ($T_{abortive}$), elongation from position +13 to position +60 ($T_{elongation}$), and the time for probe hybridization ($T_{hybridization}$). Because promoter opening by RNAP is generally very fast (~30 s$^{-1}$, **Stano et al., 2002**), $T_{open}$ is negligible. Therefore:

$$T_{abortive} = \Delta T_{on} - \left(T_{elongation} + T_{hybridization}\right)$$

Based on our measurement of the RNAP peak elongation rate (300 ± 60 nt s$^{-1}$, **Figure 3**), $T_{elongation} = (60–13)/300 = $ ~0.16 s. Based on our measurement of the F1 probe hybridization rate (6 × 10$^6$ M$^{-1}$s$^{-1}$, **Figure 1C**), $T_{hybridization} = $ ~0.33 s (at the 500 nM probe concentration used in the experiment). Thus, $T_{abortive} = 0.7 - (0.16 + 0.33) = $ ~0.2 s. This estimate of $T_{abortive}$ is within the upper limit set by our measurement of $T_{stationary} = 0.7 ± 0.3$ s (the net time RNAP spends on the DNA template without elongating, **Figure 3G**). The fact that $T_{stationary}$ is larger than $T_{abortive}$, suggests that RNAP might dwell at the end of the DNA template for ~0.5 s before run-off. Pausing/arresting at free DNA ends has been previously observed with multi-subunit bacterial and eukaryotic RNA polymerases in ensemble assays, and likely involved backtracking by these RNAPs (lifetimes of >1 min, **Arndt and Chamberlin, 1988**; **Izban et al., 1995**). It remains to be tested whether T7 RNAP dwells at the end of the DNA via a similar mechanism, and whether the pause is part of a natural termination process.

## Discussion

Here we present a strategy to design fast probe-target hybridization pairs for quantitative single-molecule detection of nascent RNA transcripts at sub-second time resolution under physiological conditions (fastFISH). The hybridization rates of fastFISH probes were near ~10$^7$ M$^{-1}$s$^{-1}$ which exceeded annealing rates of probe-target sequences reported in literature by up to three orders of magnitude. We followed three simple rules in fastFISH probe-target design. First, the targets and the probes should be unstructured, which we achieved using sequences composed of only three bases—A, U, and C for RNA targets, and A, T, and G for DNA probes. We note that the three-base rule of fast hybridization also applies to DNA–DNA pairs (data not shown). Second, a complexity filter should be applied to ensure specificity and 1-to-1 stoichiometry of hybridization. This is essential for single-molecule counting, and cannot be achieved by using repetitive sequences such as poly (A/T). Third, fastFISH probes should be self-quenched. This reduces the fluorescence background from the free probe, and enables the use of near-micromolar probe concentrations

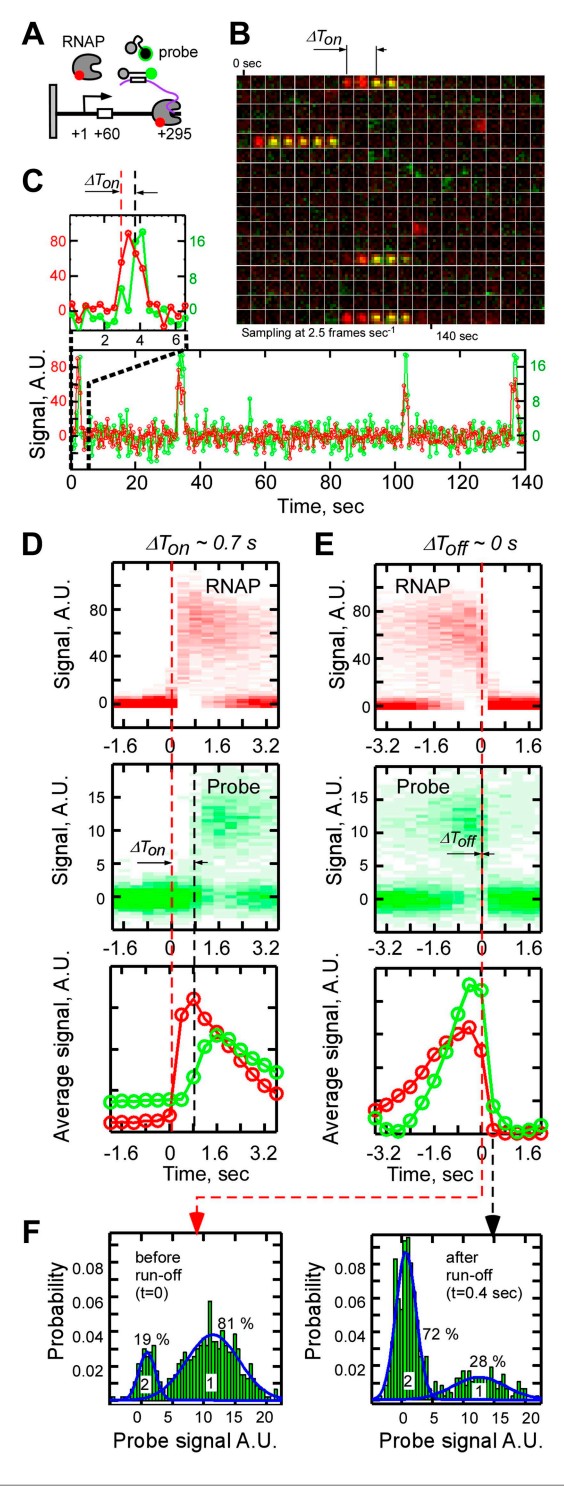

**Figure 4**. Single-molecule dissection of the T7 RNAP transcription cycle using fastFISH. (**A**) Schematic of experiment. (**B**) Merged video montages (5 × 5 pixel region of interest) centered at a single, previously photobleached DNA molecule. Interactions of RNAP (false colored in red) and probe (false colored in green) with the DNA locus were *Figure 4. Continued on next page*

for real-time single-molecule imaging. Self-quenching of fastFISH probes likely occurs through random polymer motion and/or contact quenching (*Johansson et al., 2002*; *Marras et al., 2002*), and does not rely on hairpin formation (which is known to compromise hybridization rates [*Tsourkas et al., 2003*]). The choice of probe length of ~20 nucleotides was based on the end-to-end distance of a 20-mer nucleic acid duplex, expected to separate the fluorophore and the quencher beyond typical Forster distance (4~6 nm) upon hybridization. Shorter probes in combination with a short-range quencher may be used (*Zhu et al., 2005*) if the potential faster off-rate of the probe is not a concern. We note that probe hybridization rates could be further increased using peptide nucleic acid (PNA) probes, because the neutral charge of the PNA backbone is expected to reduce the electrostatic repulsion between the probe and the target (*Kuhn et al., 2002*).

To use fastFISH, the three-base target sequence must be inserted within the context of an existing four-base (i.e., 'structured') RNA sequence, which, in principle, might sequester the fastFISH target from hybridization. However, in our experience, the presence of adjacent four-base sequences usually does not interfere with co-transcriptional RNA detection, likely because four-letter sequences become kinetically trapped within their own secondary structures. Nevertheless, the use of a secondary structure prediction tool is recommended to ensure that flanking sequences do not interfere with the fastFISH target.

As a proof of concept, we used fastFISH to dissect the full transcription cycle of the prototypical single-subunit T7 RNAP, and obtained kinetic estimates of promoter search (upper limit of non-specific RNAP-DNA interactions: $T_{search}$ ~80 ms), promoter binding, promoter escape, elongation and termination by RNAP (summarized in *Figure 5*). We recorded several differences between our measurements and previous studies. First, the measured rate of stochastic RP$_c$ dissociation ($T_0$ ~0.3 s, $k_{off}$ ~3 s$^{-1}$) was higher than what had been reported in previous ensemble and single-molecule studies ($k_{off}$ from 0.4 s$^{-1}$ to 1 s$^{-1}$ [*Tang and Patel, 2006*; *Tang et al., 2009*]). Second, the measured rate of promoter escape ($T_{abortive}$ ~0.2 s, $k_{escape}$ ~5 s$^{-1}$) was higher than in previous

*Figure 4. Continued*

imaged simultaneously at 2.5 Hz. Four representative transcriptional events are shown. (**C**) Fluorescence time traces corresponding to the montage shown in Panel **B** (RNAP and probe are shown in red and green, respectively). The first transcriptional event is zoomed in and shown on top. (**D**) Calculation of the time delay of probe binding with respect to RNAP binding ($\Delta T_{on}$) from heat maps of all RNAP-DNA (top) and all probe-DNA binding events (middle) post-synchronized by RNAP binding ($T_{on}$, $N_{DNA}$ = 112, $N_{events}$ = 469). Baseline of zero intensity indicates no RNAP/probe binding. Time point $t$ = 0 corresponds to the frame immediately before RNAP binding (red dashed line). On the bottom are the weight-averaged, normalized signal intensities of RNAP and probe binding calculated based on the heat-maps of all RNAP-DNA (red) and probe-DNA (green) binding events. (**E**) Calculation of the time delay of probe dissociation with respect to RNAP run-off ($\Delta T_{off}$) from heat maps of all RNAP-DNA and probe-DNA dissociation events post-synchronized by RNAP dissociation ($T_{off}$). Time point $t$ = 0 corresponds to the frame immediately before RNAP dissociation (red dashed line). (**F**) Calculation of the efficiency of real-time RNA detection (left), and of the fraction of events in which RNA was released from DNA upon RNAP run-off (right). Productive RNAP-DNA interactions were post-synchronized by RNAP dissociation as shown in Panel E, and probability distribution of probe signal intensity was plotted for the time point immediately before RNAP run-off (left, $t$ = 0, for the efficiency of RNA detection) and the time point immediately after RNAP run-off (right, $t$ = 0.4 s, for the fraction of RNA released upon run-off). Fits of distributions to sums of two Gaussian functions are shown in blue ($R^2$ = 0.92). The higher mean-value Gaussian (peak 1) corresponds to the events with the probe signal present and the lower mean value (peak 2) corresponds to the events without the probe signal present.

The following figure supplements are available for figure 4:

**Figure supplement 1**. Control single-molecule transcription experiments in the presence of an incomplete set of NTPs.

**Figure supplement 2**. Sequence-specificity control of real-time RNA detection by fastFISH.

**Figure supplement 3**. Effect of the distance between the promoter and the probe target sequence on the time delay between probe and RNAP binding, $\Delta T_{on}$.

ensemble and single-molecule studies ($k_{escape}$ from 0.25 s$^{-1}$ to 1 s$^{-1}$, [*Skinner et al., 2004*; *Tang et al., 2009*]). Third, the measured peak elongation rate (300 ± 60 nt s$^{-1}$) was somewhat higher than in previous ensemble measurements (~240 nt s$^{-1}$, *Bonner et al., 1994*; *Golomb and Chamberlin, 1974*), and significantly higher than in previous single-molecule measurements (from 40 nt s$^{-1}$ to 120 nt s$^{-1}$, *Thomen et al., 2008*; *Skinner et al., 2004*). The overall faster rates measured in our study are likely due to the use of optimal physiological temperatures (37°C). Indeed, the catalytic rate of the T7 RNAP measured in steady-state conditions is known to increase up to 10-fold in the 20°C–37°C temperature range (*Maslak and Martin, 1993*). Another potential contributing factor to the faster observed rates could be that our method to dissect the RNAP transcription cycle does not require the use of force, or covalent modification of the DNA templates.

We note that the dwell time distribution of productive RNAP-DNA interactions was skewed towards longer-lasting events, likely due to spontaneous RNAP pausing. Our observation of pausing might explain previously reported premature dissociation of colliding elongating RNAPs, observed at high RNAP:template ratios ('bumping', *Zhou and Martin, 2006*).

FastFISH detection of nascent RNA, coupled to single-molecule detection of protein-DNA interactions will be particularly useful to study the dynamics of initiation, escape, elongation, and termination by multi-subunit RNAPs, and the coupling of the transcriptional steps to downstream RNA-processing events. The guidelines for designing fastFISH probe-target pairs can be used to efficiently detect endogenous RNAs in situ and in vivo, to silence gene expression, and to build novel nucleic acid nanostructures.

## Materials and methods

### Prediction of secondary structure, and calculation of complexity of random oligonucleotides comprised of different alphabets

Random 19-mer DNA and RNA sequences were generated using a home-written Matlab routine. Briefly, built-in matlab routine randperm was used to pick a random set of k integers from a pool of n integers, where n = 3$^{19}$ (for 3-base sequences) or 4$^{19}$ (for 4-base sequences), and k = 3$^{12}$ (531441). A different seed for the Matlab random number generator was used in independent trials to ensure independent sampling. (Sets of 3-base 19-mers were used, because a 19-mer is representative of commonly used hybridization probes, and analysis of k = 3$^{12}$ 19-mer sequences could be performed

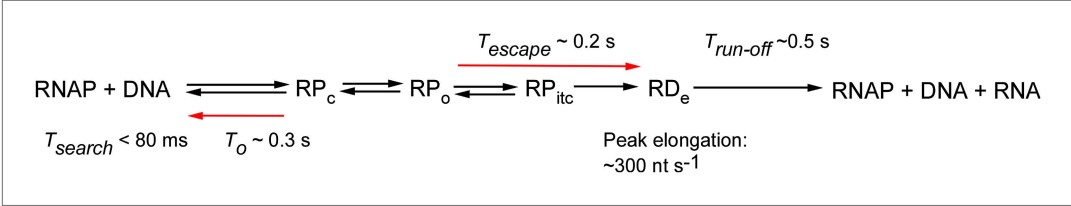

**Figure 5**. Summary of RNAP transcriptional kinetics measured in this work.

on a modern portable computer.) Built-in Matlab routine dec2base was then used to convert sets of k integers into strings of numeric characters of base 3 (0, 1, and 2—for 3-base sequences) or base 4 (0, 1, 2, and 3—for 4-base sequences). The numeric characters were then replaced with A, U, C, and G, and sequences having GC-content ≥0.4 and ≤0.6 were saved as ASCII files, to be processed with the secondary structure prediction tool Mfold. Calculation of free energies of self-folding for DNA and RNA sequences was performed on a personal computer (routine hybrid-ss-min, release 3.8, available at http://mfold.rna.albany.edu, *Zuker, 2003*). For DNA, folding parameters −'sodium = 0.15 −magnesium = 0.01' were used to match the composition of hybridization buffer used in single-molecule experiments on probe on-rate measurements. For RNA, default folding parameters were used. The free energies of self-folding were extracted from Mfold output files using a home-written Matlab parsing routine, and plotted as normalized probability histograms with 1 kcal mol$^{-1}$ binning. At least three independently randomized sets of $3^{12}$ sequences were analyzed for each alphabet (rAUC, rAUG, rCAG, rCUG, rAUGC, ATC, ATG, CAG, CTG, and ATGC) and results from independent sets were consistent to within 0.5%.

Sequence complexity was calculated using Matlab routine kolmogorov (provided by Stephen Faul, Radisens Diagnostics, code available at the Mathworks website), modified to output the absolute number of Lempel–Ziv operations, not normalized for the sequence length (*Kaspar and Schuster, 1987*). Per the modified kolmogorov routine, the complexity measure of a sequence comprised of 19 unique symbols (e.g., 'abcdefghijklmnopqrs') is 19, and the complexity measure of a sequence comprised of 19 identical symbols (e.g., $A_{19}$) is 2. To calculate the complexity of all tiling 19-mers of the human chromosome 22, the genomic sequence (length: 51,304,560 bp) and the annotation file indicating the locations of exons of known genes were downloaded from the UCSC genome browser, the sequences of exons in the forward strand were assembled into one contiguous string, and the complexities were calculated using a 19-base sliding window.

## Sequences of fastFISH probes and their variants

The sequences of the F1 probes were as follows (synthesized by Integrated DNA technologies, IDT, Coralville, IA):

F1: 5'-Cy3-GTT AAG ATA AGG GAT AGG G-3' (not self-quenched);
F1: 5'-Cy3-GTT AAG ATA AGG GAT AGG G-FQ-3' (self-quenched with Iowa Black FQ)
F1: 5'-Atto633-GTT AAG ATA AGG GAT AGG G-RQ-3' (self-quenched with Iowa Black RQ)

The sequences of self-quenched F2 probes and the F2+1C, F2+2C, F2+3C, and F2+4C variants (these variants were selected as representative of their categories based on the free energies of self-folding) were as follows (synthesized by Fidelity Systems, Gaithersburg, MD):

F2: 5'-BHQ-3-GGT GTA TGT AAT TGG AGT GGT T-C6-Atto633-3';
F2+1C: 5'-BHQ-3-GGT GTA TCT AAT TGG AGT GGT T-C6-Atto633-3';
F2+2C: 5'-BHQ-3-GGT GTA TGC AAT TGG ACT GGT T-C6-Atto633-3';
F2+3C: 5'-BHQ-3-GGT GTA TGT AAC CGG AGT GCT T-C6-Atto633-3';
F2+4C: 5'-BHQ-3-GCT GTA TGC AAT TGG AGC CGT T-C6-Atto633-3'.

In the above sequences, BHQ-3 is Black Hole 3 quencher, C6 is a six-carbon linker.
The sequences of RNA targets used in single-molecule TIRF/CoSMoS measurements of probe hybridization rates were as follows:

F1F2: 5'- <u>CCC UAU CCC UUA UCU U</u>*AA CCA CUC CAA UUA CAU ACA C*CC UUC AAA CUU CAA ACU UCA AAG CAC AAG UUU UAU CCG GCC UUU AUU CAC AUU-bio-3'

F1F2+1G: 5'- <u>*CCC UAU CCC UUA UCU U*</u>*AA CCA CUC CAA UUA **G**AU ACA CCC* UUC AAA CUU CAA ACU UCA AAG CAC AAG UUU UAU CCG GCC UUU AUU CAC AUU-bio-3'

F1F2+2G: 5'- <u>*CCC UAU CCC UUA UCU UAA CCA*</u> *GUC CAA UU**G** CAU ACA CCC* UUC AAA CUU CAA ACU UCA AAG CAC AAG UUU UAU CCG GCC UUU AUU CAC AUU-bio-3'

F1F2+3G:5'- <u>*CCC UAU CCC UUA UCU UAA G*</u>*CA CUC C**G**G UUA CAU ACA CCC* UUC AAA CUU CAAACU UCA AAG CAC AAG UUU UAU CCG GCC UUU AUU CAC AUU-bio-3'

F1F2+4G:5'- <u>*CCC UAU CCC UUA UCU UAA C***G**G *CUC CAA UU**G** CAU ACA **G**CC* UUC AAA CUU CAA ACU UCA AAG CAC AAG UUU UAU CCG GCC UUU AUU CAC AUU-bio-3'

In the above sequences, regions that correspond to the F1 and F2 targets are underlined and italicized, respectively, and the variations within the F2 target region to introduce 1-4 G residues are highlighted in bold. Note that the F1 and F2 targets overlap by three bases. The biotin residue on the 3' end enabled immobilization of the RNA targets on surface for single-molecule imaging by TIRF/CoSMoS. To fluorescently label the RNA targets, a Cy3-labeled DNA oligonucleotide (sequence: 5'-Cy3-AAT GTG AAT AAA GGC CGG ATA AAA CTT GTG C-3' IDT) was annealed to the 3' of the RNA targets.

## Single-molecule imaging instrument

Single-molecule measurements of probe-target hybridization rates and T7 RNAP transcription were performed using a home-built multi-color total-internal-reflection microscope as described (*Revyakin et al., 2012*). In brief, fluorescence was excited and imaged through a 60 × 1.49 NA objective lens (Olympus, Center Valley, PA). The objective lens and the sample holder were pre-heated to 37°C (Bioptechs, Butler, PA). Fluorescence was excited with 532 nm (Coherent Verdi G2, intensity 100–300 W/cm$^2$, Coherent, Santa Clara, CA) and 640 nm (Coherent Cube 100, intensity 100 W/cm$^2$) lasers, split into two imaging channels (band centered at 580 nm, width 60 nm; and band centered at 675 nm, width 45 nm), and imaged using two separate, synchronized EMCCD cameras (512 × 512 pixels, 1 pixel = 200 nm, conventional acquisition at 2.5 Hz or electron-multiplied acquisition at 12.5 Hz) (Andor, Belfast, United Kingdom). The position of the sample with respect to the objective lens was actively stabilized in *(x,y,z)*. To that end, 2.8 µm-diameter magnetic beads (Life Technologies, Carlsbad, CA) decorating the imaging surface were illuminated with an infrared (IR) light-emitting diode (851 nm, Roithner LaserTechnik, Vienna, Austria), tracked with an IR camera using real-time image-processing (*Gosse and Croquette, 2002*; *Revyakin et al., 2012*) at 30 Hz, and the position of the bead in the field of view was adjusted in real-time at 1–5 Hz using a 3-axes nanopositioning stage (Physik Instrumente, Karlsruhe, Germany). The IR camera and framegrabber for bead tracking were from 1stVision (Andover, MA).

## Preparation of glass flow cells for single-molecule measurements of probe-target hybridization rates

Unless otherwise noted, all chemicals were purchased from Sigma-Aldrich (St. Louis, MO). Borosilicate coverslips were placed in ceramic racks (Thomas Scientific, Swedesboro, NJ), cleaned with piranha solution (a mixture of three parts concentrated $H_2SO_4$ and one part 30% $H_2O_2$, which is extremely corrosive and explosive) twice for 30 min, rinsed copiously with deionized water, and hydroxylated with 0.5 M KOH solution for 1 hr with sonication. The coverslips were then rinsed copiously with deionized water, rinsed twice by dipping in acetone (Chromasolv-Plus) for 10 s, and placed into a 3% solution of aminopropyltriethoxylane (APTES, Thermo Fischer Scientific, Waltham, MA) in acetone for 45 min, with gentle shaking. APTES-treated coverslips were then rinsed copiously with deionized water, and blow-dried with nitrogen. 50 µl drops of solution of 10% PEG-succinimidyl valerate (PEG-SVA, Mw = 5,000, Laysan) and 0.05% biotin-PEG-SVA (Mw = 5,000, Laysan) in 0.45M $K_2SO_4$ and 0.1M $NaHCO_3$ (pH 9.0) was squeezed between pairs of coverslips, and the coverslip pairs were incubated with PEG at 30°C for 30 min. PEG-treated coverslips were then rinsed copiously with deionized water, blow-dried with nitrogen, and the unreacted amine groups were end-capped by squeezing a 50 µl drop of solution of 2 mg sulfo-succinimidyl acetate (Pierce) in 0.1M $NaHCO_3$ (Pierce) for 10 min. The PEG-treated and amine-capped coverslips were rinsed copiously with deionized water, blow-dried with nitrogen, and stored dry at −80°C. (We found that end-capping of unreacted APTES amine groups was essential to eliminate non-specific interactions of fluorescently labeled probes with the glass surfaces at high probe concentrations.) An imaging flow cell containing seven reaction channels (volume of each channel: ~25 µl) and side injection ports was constructed using two PEG-treated coverslips and double-sided adhesive tape (3M VHB 4095). The flow cell was mounted on the microscope, and the inner surface was decorated with the 2.8 µm magnetic beads for active stage stabilization (~1–3 beads per 100 × 100 micron field of

view) and functionalized with streptavidin (by incubating 5 µg/ml solution of streptavidin in phosphate-buffered saline for 30 s in a channel immediately before use).

## Measurement of probe-target hybridization rates with TIRF/CoSMoS

Biotinylated, fluorescently labeled synthetic RNA targets were captured in the flow cell at a density ~1000–2000 molecules per field of view (~100 × 100 µm) by injecting 20 pM solution of RNA target in loading buffer (phosphate-buffered saline containing 0.08% Tween-20). The position of the microscope stage was then actively stabilized in *(x,y,z)* (see above), and a 'mapping' video of diffraction-limited spots indicating the positions of RNA target molecules was acquired at 2.5 Hz until all RNA spots were photobleached. Single-molecule imaging here and below was conducted in the presence of 0.9 mM Trolox, 2.2 mM protocatechuic acid (PCA), and 10 µg/ml protocatechuate dehydrogenase (Toyobo), further purified to remove contaminating nuclease activity (Jonas Korlach, Pacific Biosciences, personal communication). After mapping of the RNA targets, the loading buffer was replaced with hybridization buffer (loading buffer plus 10 mM $MgCl_2$) and a solution of fluorescently labeled, self-quenched probes in hybridization buffer at concentrations in the 2–500 nM range was injected into the flow cell at flow rate ~50 µl s$^{-1}$. A real-time video of probe binding events (observed as appearance of diffraction-limited fluorescent spots, *Figure 1—figure supplement 2*) was acquired at 2.5 Hz. For most accurate measurements of the hybridization rates, the concentrations of probes were adjusted empirically to achieve average probe arrival times of 10–100 s. Data analysis was essentially as described (*Revyakin et al., 2012*). Briefly, single-molecule spots were identified in every frame of the target and probe videos using the software Insight (*Huang et al., 2010*), and each spot was localized in *(x,y)* by fitting a 5 × 5 pixel spot intensity distribution to a 2D Gaussian. The *(x,y)* coordinates of all identified spots in all video frames were grouped into clusters ('blobs') using a 2D histogram (1 pixel binning), and the average *(x,y)* of every blob (indicating the locations of target and probe molecules) was calculated. Then, for every target blob, the closest probe blob was found, the deviations *ΔxΔy* were calculated for all such probe-target pairs, and plotted as a 2D histogram ('the co-localization plot'). RNA targets that were located within two standard deviations of the central peak at position (0,0) of the co-localization plot (~40 nm) were selected. Such 'active' RNA molecules typically comprised ~30% of all RNA molecules. Time series of probe-target binding was generated for every active RNA (using the mean fluorescence intensity from a 5 × 5 pixel region of interest (ROI), with the mean intensity of the ROI perimeter pixels subtracted as background). Each time series was examined, and the video frame corresponding to the probe arrival at the target was identified (*Figure 1—figure supplement 2*). The dataset containing all probe arrival times ($T_{wait}$, usually 100–500 events) was binned, with the center of the first bin corresponding to the minimal observed $T_{wait}$, the center of the last bin corresponding to the maximal observed $T_{wait}$, and the number of bins $N_{bins} = \sqrt[3]{(2N_{events})}$ (*Scott, 1992*). The first bin was discarded to account for the discontinuity, and for the lag time due to the buffer exchange during probe injection. The probability histogram of $T_{wait}$ (*Figure 1—figure supplement 2*) was fit to a single exponential function using the Matlab built-in routine nlinfit, which gave an estimate of the characteristic hybridization time (95% confidence interval), and the goodness of fit (typically, $R^2 > 0.98$). On-rates for probes F1, F2, F2+1C, F2+2C, F2+3C and F2+4C were calculated as $k_{on} = 1/(T_{wait} \times [probe])$.

## Promoter DNA constructs for single-molecule transcription assays

Promoter DNA constructs were amplified by PCR. To attach and image the DNA constructs, the PCR primer complementary to the sequence upstream from the promoter was labeled with biotin and Cy3/Cy5/Atto633 at the 5'-end. In most experiments, the PCR-amplified DNA template spanned from −75 to +295 with respect to the transcription start site (+1). In the DNA template sequence shown below, the underlined sequences are, respectively: the upstream PCR primer (5' base at −75), the consensus T7 RNAP promoter, the F1 probe target, the F2 probe target, and the downstream PCR primer (5' base of the complementary sequence at +295). The +1 base is shown in bold.

(−75) <u>TTATGTATCATACACAT</u>ACGATTTAGGTGACACTATAGAACTCGAGCAGCTGGATCC
<u>T A A T A C G A C T C A C T A T A</u> **G** G G A G A C C A C A A C G G T T T C C C T C T A G A
<u>C C C T A T C C C T T A T C T T A A C</u>G A A T T G T G A G C G C T C A C A A T T C A A A C T T T C A A A C T T C A A A
C T T C A A A C T T C A A A C T T C A A A C T T C A A A C T T C A A A C T T C A A A C T T C A A A C T T C A A
A C T T C A A A C T T C A A A C T T C A A A C T T G A A T T C T T T C A A A A<u>C A C T C C A A T T A C A T A C A C C</u>
T T T C A A A A C C A C C G T T G A T A T A T C C C A A T G G C T G C A G C T G G A T A T T A C G G C C T T
T T T A A A G A C C G T A A A G A A A A A T A A<u>G C A C A A G T T T T A T C C G G C</u> (+295)

For PCR-amplify DNA templates containing longer transcribed sequences (ending at +633 and +910), the same upstream primer, and primers with 5′ ends corresponding to +633 and +910 were used. The sequence for the +910 DNA template is shown below, with the regions corresponding to the +633 and +910 primers underlined, and the sequence upstream from +296 identical to the sequence of the −75…+295 template:

(+296) CTTTATTCACATTCTTGCCCGCCTGATGAATGCTCATCCGGAATTCCGTATGGCAA

TGAAAGACGGTGAGCTGGTGATATGGGATAGTGTTCACCCTTGTTACACCGTTTT
CCATGAGCAAACTGAAACGTTTTCATCGCTCTGGAGTGAATACCACGACGATTTCCGGC
AGTTTCTACACATATATTCGCAAGATGTGGCGTGTTACGGTGAAAACCTGGCCTATTT
CCCTAAAGGGTTTATTGAGAATATGTTTTTCGTCTCAGCCAATCCCTGGGTGAGTTTC
ACCAGTTTTGATTTAAACGTGGCCAATAT<u>GGACAACTTCTTCGCCCCCGTTTT</u>
CACCATGGGCAAATATTATACGCAAGGCGACAAGGTGCTGATGCCGCTGGCGATTC
AGGTTCATCATGCCGTCTGTGATGGCTTCCATGTCGGCAGAATGCTTAAT
GAATTACAACAGTACTGCGATGAGTGGCAGGGCGGGGCGTAATTTTTTT
AAGGCAGTTATTGGTGCCCTTAAACGCCTGGTGCTACGCCTGAATAAGTGATA
ATAAGCGGATGAATGGCAGAAATTCGCCGGATCTTTGTGAA<u>GGAACCTTA</u>
<u>CTTCTGTGGTGTGACATA</u> (+910)

In the DNA template containing the null T7 RNAP promoter mutation (experiments in *Figures 2 and 3*), the wild type promoter sequence was replaced with TAA TA<u>A CC</u> ACT CAC TAT AGG G (the mutation is underlined). In the DNA template containing deleted F1 target (experiment in *Figure 4— figure supplement 2A*), the sequence of the F1 target was replaced with CAA ACT TCA AAC TTC AAA C. In the DNA template containing deleted F2 probe target (experiment in *Figure 4—figure supplement 2B*), the sequence of the F2 target was replaced with CAA ACT TCA AAC TTC AAA C.

We note that, in the context of the full transcript, the 3-base unstructured RNA target sequences might be potentially trapped in secondary structures formed by 4-base sequences present in the same transcript. Therefore, we used Vienna RNA-folding software (*Gruber et al., 2008*) to calculate the secondary structure of the full transcripts to ensure that 4-base sequences do not hybridize to the 3-base targets, and, instead, tend to self-hybridize locally.

## Synthesis of Cy5-PEG$_4$-HaloTag Ligand (S3)

Commercial reagents were obtained from reputable suppliers and used as received. All solvents were purchased in septum-sealed bottles stored under an inert atmosphere. Reactions were monitored by thin layer chromatography (TLC) on precoated TLC glass plates (silica gel 60 F$_{254}$, 250 µm thickness) or by LC/MS (4.6 mm × 150 mm 5 µm C18 column; 5 µl injection; 10–95% or 50–95% CH$_3$CN/H$_2$O, linear gradient, with constant 0.1% vol/vol TFA additive; 20 min run; 1 ml/min flow; ESI; positive ion mode; UV detection at 254 nm). High-resolution mass spectrometry was performed by the Mass Spectrometry Center in the Department of Medicinal Chemistry at the University of Washington. NMR spectra were recorded on a 400 MHz spectrometer. $^1$H and $^{13}$C chemical shifts (δ) were referenced to TMS or residual solvent peaks, and $^{19}$F chemical shifts (δ) were referenced to CFCl$_3$. Data for $^1$H NMR spectra are reported as follows: chemical shift (δ ppm), multiplicity (s = singlet, d = doublet, t = triplet, q = quartet, dd = doublet of doublets, m = multiplet), coupling constant (Hz), integration.

## Cy5-PEG$_4$-HaloTag Ligand (S3)

Cyanine dye **S1** (10 mg, 13.0 µmol, 1.33 eq; prepared as described in (*Mujumdar et al., 1993*)) was combined with *N,N′*-disuccinimidyl carbonate (3.3 mg, 13.0 µmol, 1.33 eq) and DMAP (0.16 mg,

1.30 µmol, 0.133 eq) in DMF (1 ml), and triethylamine (7.2 µl, 51.9 µmol, 5.33 eq) was added. The reaction was stirred at room temperature for 1 hr, at which point LC/MS analysis indicated complete conversion to the NHS ester. HaloTag ligand (O4)amine **S2** (3.4 mg, 9.76 µmol, 1 eq) in DMF (0.5 ml) was then added, and the reaction was stirred at room temperature for 18 hr while shielded from light. The crude reaction mixture was directly purified by reverse phase HPLC (10–95% MeCN/H$_2$O, with constant 0.1% vol/vol TFA additive) to provide **S3** (6.9 mg, 66%, TFA salt) as a blue-purple solid. $^1$H NMR (MeOD, 400 MHz) δ 8.318 (t, $J$ = 13.0 Hz, 1H), 8.313 (t, $J$ = 13.0 Hz, 1H), 7.97–7.85 (m, 4H), 7.34 (dd, $J$ = 8.1, 2.3 Hz, 2H), 6.68 (t, $J$ = 12.5 Hz, 1H), 6.35 (d, $J$ = 13.6 Hz, 1H), 6.34 (d, $J$ = 13.6 Hz, 1H), 4.24–4.07 (m, 4H), 3.66–3.49 (m, 16H), 3.45 (t, $J$ = 6.5 Hz, 2H), 3.36–3.33 (m, 2H), 2.22 (t, $J$ = 7.3 Hz, 2H), 1.93–1.63 (m, 6H), 1.76 (s, 12H), 1.57 (p, $J$ = 6.9 Hz, 2H), 1.52–1.23 (m, 9H); $^{19}$F NMR (MeOD, 376 MHz) δ −75.66 (s); Analytical HPLC: >99% purity (4.6 mm × 150 mm 5 µm C18 column; 5 µl injection; 10–95% CH$_3$CN/H$_2$O, linear gradient, with constant 0.1% vol/vol TFA additive; 20 min run; 1 ml/min flow; ESI; positive ion mode; detection at 254/633 nm); HRMS (ESI) calculated for C$_{47}$H$_{69}$ClN$_3$O$_{11}$S$_2$ (M)$^+$ 950.4062, found 950.4078.

## Purification and labeling of T7 RNAP

T7 RNAP was purified as recombinant, N-terminal 6xHis and 6xHis-Halo fusions from the *E. coli* strain BL21 (DE3) using standard Ni-NTA chromatography per recommendations of the Ni-NTA resin manufacturer (Qiagen, Hilden, Germany). Purified proteins were dialyzed against storage buffer (contains 20 mM potassium phosphate pH 7.5, 100 mM NaCl, 50% glycerol, 10 mM DTT, 0.1 mM EDTA and 0.2% sodium azide) and stored at −20°C. To label Halo-RNAP, 20 µM of protein was mixed with 60 µM Cy5-PEG4-HaloLigand in reaction buffer (contains 50 mM HEPES pH 7.5, 150 mM NaCl, 0.5 mM EDTA, 1 mM DTT, and 0.005% NP40), incubated at room temperature for 1 hr, and then at 4°C overnight. To remove unreacted HaloLigand, the reaction mixture was passed through a PD-10 column (GE Healthcare, Waukesha, WI) pre-equilibrated with the storage buffer. Protein labeling efficiencies were >90% based on UV-Vis absorption spectroscopy. Proteins were examined using SDS-polyacrylamide gel electrophoresis and analysis of Cy5 fluorescence with Typhoon Trio+, (GE Healthcare), followed by Coomassie Brilliant Blue staining.

## Ensemble transcription by T7 RNAP

All transcription reaction mixture contained the following ingredients unless otherwise specified: 40 mM Tris–HCl pH 7.9, 6 mM MgCl$_2$, 2 mM Spermidine, 0.5 mM each nucleotide triphosphate (NTP), 4 ng/µl yeast tRNA (Sigma, further purified with protease K treatment and phenol/chloroform extraction), 100 ng/µl bovine serum albumin (New England Biolabs, Beverly, MA), 2 µM carrier DNA oligonucleotide (sequence: 5′-GTA TTG AGT CTT CAT TCT GTA T-3′, IDT), 0.08% Tween 20 (EMD Millipore, Gibbstown, NJ), 0.9 mM Trolox, 2.2 mM PCA, and 0.3 U/µl RNasin (Promega Madison, WI). In ensemble measurements, 10 nM DNA template (same PCR fragment as used in single-molecule assays), 0.17 µCi/µl α-$^{32}$P-ATP (Perkin Elmer, Waltham, MA) and different concentrations of RNAP were also included in the reaction. After incubation for 10 min at 37°C, 2 µl of the transcription reaction was loaded onto a 6% denaturing polyacrylamide gel. The gel was dried under vacuum at 80°C for 1 hr, exposed to a PhosphorImager screen and scanned (Typhoon Trio+, GE Healthcare).

## Single-molecule transcription by T7 RNAP

The protocol for preparation of passivized glass surfaces for single-molecule transcription measurements will be described in detail elsewhere. Briefly, borosilicate coverslips (VWR) were cleaned with piranha solution as described above, blow-dried with nitrogen, and spin-coated to create a 50-nm layer of polystyrene containing 1% azide-terminated polystyrene on the glass surface. An imaging flow cell was assembled as described above, and the imaging surface was functionalized with biotin-PEG-alkyne as described (*Presolski et al., 2011*). The flow cell was then mounted on the microscope holder, and the objective lens and the flow cell were pre-heated to 37°C. The imaging surface was blocked by incubation with loading buffer (phosphate-buffered saline containing 0.08% Tween-20) for 5 min, and functionalized by incubation with the loading buffer containing 5 µm/ml streptavidin for 30 s. Biotinylated, fluorescently labeled DNA templates were captured and 'mapped' in the field of view of the microscope as described for the probe-target hybridization experiments. The position of the microscope stage was actively stabilized in *(x,y,z)*, and 100 µl of transcription reaction mixture (see above) supplemented with 10 µg/µl protocatechuate dehydrogenase, 5 nM Cy5-Halo-labeled or (His)6 tagged, unlabeled RNAP, and/or 500 nM of fastFISH probe was injected into the flow cell. Videos of real-time interactions of RNAP and/or probe with the mapped, photobleached DNA loci were

recorded with synchronized EMCCD cameras at 2.5 Hz, unless otherwise specified. Single-molecule transcription reactions typically lasted for 15 min.

## Analysis of RNAP and probe interactions with active DNA loci

Diffraction-limited RNAP spots were identified in every video frame, localized by 2D Gaussian fitting, and co-localized with the previously mapped DNA loci as described above for the probe-target hybridization experiments. Time series of RNAP and probe fluorescence intensities were generated for every 'active' DNA molecule using an ROI of 3 × 3 pixels, with 1-pixel perimeter background subtraction. A DNA locus was defined as 'active' if it co-localized with an RNAP spot in 5–9 cumulative frames (user-defined parameter) during the 15-min reaction. Videos frames in which RNAP bound to DNA ($T_{on}$) and dissociated from DNA ($T_{off}$) were identified in the time series manually, and the dwell times of RNAP-DNA interactions ($T_{off} - T_{on}$) were calculated. The probability distributions of RNAP dwell times were fit to a single exponential function, or to a sum of single exponential and Gaussian functions using Matlab routine nlinfit.

To measure the average time delay between binding of RNAP and the probe during productive RNAP-DNA interactions ($\Delta T_{on}$), RNAP-DNA interactions that had dwell times ≥0.8 s were selected, post-synchronized by their $T_{on}$, and a two-dimensional histogram of all RNAP binding events (a heat map of signal intensity vs frame number [*Figure 4*, and *Figure 4—figure supplements 1–3*]) was generated (routine histcn, Bruno Luong, Fogale Nanotech, code available on the Mathworks website). The same dataset of RNAP $T_{on}$ was then used as a reference to calculate a heat map of all probe binding events in the respective synchronized probe videos. For every frame of the heat map, the centers of mass of RNAP and probe signal intensity distributions were calculated, and normalized by setting the maximum intensity to 1, and weight-averaged profiles of all RNAP and probe binding events were generated. The rising edges of the averaged RNAP and probe profiles were found by fitting the maxima of the respective first derivatives to second-degree polynomial, and $\Delta T_{on}$ was calculated by subtracting the edge for probe from the edge for RNAP. The same approach was used to calculate $\Delta T_{off}$ (*Figure 4E*), except that video frames in which RNAP dissociated from the DNA ($T_{off}$) were used as references to post-synchronize RNAP- and probe-DNA dissociation events to generate respective heat maps, and to calculate weight-averaged profiles of all RNAP and probe dissociation events.

To calculate the efficiency of nascent RNA detection by the probe (*Figure 4F*), the probe dissociation heat map was used to calculate the distribution of the probe signal intensity at the last frame before RNAP dissociation. The distribution was then fit to a sum of two Gaussians using the Matlab routine nlinfit, and the ratio of the areas under the two Gaussians was calculated to give the percentage of productive RNAP-DNA interactions that contained the probe before run-off (i.e., the efficiency of RNA detection).

To calculate the fraction of RNAP run-off events in which the RNA remained at the DNA locus after RNAP dissociation, the probe dissociation heat map was used to calculate the distribution of the probe signal intensity in the frame immediately after RNAP dissociation (+0.4 s). The distribution was then fit to a sum of two Gaussians, and the ratio of the areas under the two Gaussians was calculated.

To assess the decay of RNAP fluorescence signal during elongation out of the TIRF evanescent field (*Figure 3H*), the weight-averaged profiles of all RNAP binding and dissociation events (obtained for templates having transcribed segments of different lengths—295, 633, and 910 bp) were plotted, and the edges of the RNAP dissociation profiles were manually offset in time from the edges of the RNAP binding profiles by the nearest-integer number of frames corresponding to the peak dwell time for the respective templates (*Figure 3F*).

## Acknowledgements

We thank Yan Li for molecular cloning of the designed DNA templates and technical assistance; Brian English for suggestions on data analysis; Bo Huang, Stephen Faul, and Bruno Luong for software for data analysis; Sean Eddy, Don Rio, and William Dynan for critical reading of the manuscript; and Sarah Moorehead for administrative support. Robert Tjian is a Howard Hughes Medical Institute Investigator.

## Additional information

### Competing interests

RT: Robert Tjian is President of the Howard Hughes Medical Institute (2009–present), one of the three founding funders of *eLife*. The other authors declare that no competing interests exist.

## Funding

| Funder | Author |
| --- | --- |
| Howard Hughes Medical Institute | Luke D Lavis, Robert Tjian |

The funder had no role in study design, data collection and interpretation, or the decision to submit the work for publication.

## Author contributions

ZZ, AR, Conception and design, Acquisition of data, Analysis and interpretation of data, Drafting or revising the article, Contributed unpublished essential data or reagents; JBG, LDL, Analysis and interpretation of data, Drafting or revising the article, Contributed unpublished essential data or reagents; RT, Conception and design, Drafting or revising the article

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
