## [Decision Letter]

Thank you for sending your work entitled “Single-molecule tracking of the transcription cycle by sub-second RNA detection” for consideration at *eLife*. Your article has been favorably evaluated by a Senior editor and 3 reviewers, one of whom, Jim Kadonaga, is a member of our Board of Reviewing Editors.

The Reviewing editor and the other reviewers discussed their comments before we reached this decision, and the Reviewing editor has assembled the following comments to help you prepare a revised submission.

This is an outstanding paper that describes a new single-molecule method termed fastFISH, which allows the rapid annealing of a probe to a target mRNA so that the production of the mRNA can be detected with high confidence with minimal false negatives. fastFISH will be useful for the single-molecule analysis not only of general transcription machineries but also of other RNA modifying enzymes such as poly-adenylation enzymes and poly-uridylation enzymes. The amount of work that went into the design of the fastFISH probes is impressive, and the general idea of removing internal structures with individual oligos to accelerate annealing, and in particular using three nucleotides only, will also have a broad appeal.

The technical quality of the work is high, and the data are clear. The biological insight afforded by the approach, in this study, was limited because T7 RNA polymerase has been studied extensively. Nevertheless, the direct observation of various stages of transcription starting from RNAP binding until its dissociation from the transcript has established important benchmarks for future studies of such systems and point toward exciting future possibilities of more complex systems including those with RNA polymerase II. The fastFISH approach was absolutely necessary for this work because the overall duration of the reaction was about 1–2 s and any slower annealing would have missed the majority of events.

This work is appropriate for publication in *eLife*. There are a number of minor comments that should be addressed by the authors.

Minor comments (in no particular order):

1) It is clear from the data that their probe design makes the DNA annealing 100 fold faster for probes about 20 nt long. But could they have obtained similarly rapid annealing by simply picking shorter probes, say 12 nt long? Of course, the bound state would have a shorter lifetime but it may still be long enough.

2) In Figure 2, it is difficult to distinguish between the DNA template and the newly-synthesized RNA. It might be useful to draw the RNA in a different color than the DNA.

3) In the description of Figure 3, it is stated that the long lived events encompass elongation because such events were not observed in the absence of NTPs. This statement is premature since abortive initiation was not addressed. At this stage of the paper, the long lived complexes might have derived from abortive initiation. Conceivably the apparent stability of this complex might also be a function of the DNA template length due to a cage effect. It is clear from Figure 4—figure supplement 2 that the complex is long lived; perhaps this information could be included and discussed in Figure 3.

4) It should be clearly stated in the main text that the T7RNAP used for most of the experiments is a fusion protein. I believe the halo tag is ∼34 kDa.

5) Figure 3—figure supplement 1. This Figure is supposed to show that the fluorophore tagged T7RNAP is functional. From Methods, it appears the RNAP to DNA template ratio is 1:2. I have some doubt that the assay is linear with respect to enzyme concentration. This data needs to be presented as a titration. It would also be good to include the unlabeled halo tagged RNAP in this figure to see if the N-terminal tag itself has some detrimental effect.

6) It is known (published) that bacterial and eukaryotic polymerases are slow to release from the ends of DNA during runoff transcription. This may also be true for T7RNAP. If such results are published, it would be best to cite the work on this phenomenon to avoid the appearance of inventing an ad hoc explanation for the data.

7) “Hybridization of two short NA fragments, in the simplest case, is a two-step process that begins with the unfolding of the ss fragments into a random polymer conformation, followed by complementary base-pairing.”

Random sequences are typically much more dynamic than this sentence suggests, as evidenced by the fact that the unfolding force for random ss sequences is on the order of 5 pN, threefold lower than the unzipping force of structured NA such as hairpins.

8) “Consistent with this notion, NA sequences with intrinsically higher self-folding energies have faster hybridization rates.”

This is confusing and is possibly just mis-written. Taken to the extreme it is obviously untrue. A long, perfectly matched hairpin will essentially not melt spontaneously and allow for strand invasion. Please clarify.

9) “Likewise, decreasing the length of hairpins in conventional MB increases their hybridization rate.”

Decreasing the length of a hairpin lowers its self-folding energy, so this is in contradiction with the previous statement. Again, please clarify.

10) Overall this discussion does not do a great job of separating the discussion of rate-limiting effects (unfolding vs base-pairing with target...)

11) −1.6 +/− 1.7; is the latter number the SD?

12) The description of the complexity index is not straightforward to understand (what does “by cutting and insertion” mean exactly; cutting of what, where, and insertion of what?).

13) Could the authors specify the rate of initiation of T7 RNAP in units of M^-1^ s^-1^ rather than just s^-1^?

14) Are the regions up and downstream of the hybridization target for the DNA probe also unstructured?

---

## [Author Response]

*1) It is clear from the data that their probe design makes the DNA annealing 100 fold faster for probes about 20 nt long. But could they have obtained similarly rapid annealing by simply picking shorter probes, say 12 nt long? Of course, the bound state would have a shorter lifetime but it may still be long enough*.

We agree that similarly rapid annealing can be obtained using shorter probes, as long as the target sequence in the transcript is not sequestered by secondary structure. Thus, we believe that the three-letter rule is important for selection of hybridization pairs, regardless of the probe length. In addition, as the reviewers point out, we were concerned about fast off-rates of short probes, particularly at the temperature optimal for transcriptional activity. Thus we added two sentences in the first paragraph of the Discussion section: “The choice of probe length of ∼20 nucleotides was based on the end-to-end distance of a 20-mer nucleic acid duplex (∼6.8 nm), expected to separate the fluorophore and the quencher out of the range of resonance energy transfer upon hybridization. Shorter probes in combination with a short-range quencher may be used (65) if the potential faster off-rate of the probe is not a concern.”

In the same paragraph, we also removed the last sentence on the potential use of super quenchers to further reduce fluorescence background at higher probe concentration, since this was not supported by our latest result with probes containing three quencher moieties (probably because the current self-quenching is already very efficient).

*2) In*
Figure 2*, it is difficult to distinguish between the DNA template and the newly-synthesized RNA. It might be useful to draw the RNA in a different color than the DNA*.

We have re-drawn the RNA in all schematics with a purple color.

*3) In the description of*
Figure 3*, it is stated that the long lived events encompass elongation because such events were not observed in the absence of NTPs. This statement is premature since abortive initiation was not addressed. At this stage of the paper, the long lived complexes might have derived from abortive initiation. Conceivably the apparent stability of this complex might also be a function of the DNA template length due to a cage effect. It is clear from*
Figure 4—figure supplement 2
*that the complex is long lived; perhaps this information could be included and discussed in*
Figure 3.

We have revised the second half of the first paragraph under the subtitle “Single-molecule dynamics of T7 RNAP-DNA interactions” and put more emphasis on the evidence of productive elongation in the presence of NTP. We believe that any caging effect was unlikely to play a role in these experiments, because (i) the duration of non-specific T7 RNAP-DNA interaction was very short (non-detectable at our time resolution ∼80 ms), and (ii) caging cannot explain the decrease of the fluorescence signal from the label on the T7 RNAP molecule towards the end of each individual long-lived interaction.

*4) It should be clearly stated in the main text that the T7RNAP used for most of the experiments is a fusion protein. I believe the halo tag is ∼34 kDa*.

We now state in the text that the HaloTag approach was used to label T7 RNAP.

*5)*
Figure 3—figure supplement 1*. This Figure is supposed to show that the fluorophore tagged T7RNAP is functional. From Methods, it appears the RNAP to DNA template ratio is 1:2. I have some doubt that the assay is linear with respect to enzyme concentration. This data needs to be presented as a titration. It would also be good to include the unlabeled halo tagged RNAP in this figure to see if the N-terminal tag itself has some detrimental effect*.

As suggested by the reviewers, we have performed a titration of transcriptional activity for different RNAP variants, including the Halo-tagged RNAP before labeling. Figure 3—figure supplement 1 is updated accordingly. In brief, we found that introduction of the HaloTag and labeling had no effect on transcriptional activity of the polymerase.

*6) It is known (published) that bacterial and eukaryotic polymerases are slow to release from the ends of DNA during runoff transcription. This may also be true for T7RNAP. If such results are published, it would be best to cite the work on this phenomenon to avoid the appearance of inventing an ad hoc explanation for the data*.

We have now included two references documenting formation of long-lived, arrested ternary complexes of bacterial and eukaryotic RNAP at the end of linear DNA templates during run-off. We are not aware of published biochemical work documenting formation of long-lived, arrested T7 RNAP ternary complexes at the ends of linear DNA templates during run-off. Such end-arrested complexes would be difficult to ‘catch’ biochemically: our data puts the upper limit on their lifetime at 0.5 s. The following sentences were added to the end of the Results section: “Pausing/arresting at free DNA ends has been previously observed with multi-subunit bacterial and eukaryotic RNA polymerases in ensemble assays, and likely involved backtracking by these RNAPs (lifetimes of >1 min, [1]; [23]). It remains to be tested whether T7 RNAP dwells at the end of the DNA via a similar mechanism, and whether the pause is part of a natural termination process.”

*7–10) […] Overall this discussion does not do a great job of separating the discussion of rate-limiting effects (unfolding vs base-pairing with target...*)

Questions 8-11 are closely related and we revised the first paragraph of the Results section accordingly. First, we agree that hybridization of two random sequences likely proceeds through several intermediates. We have now simplified the process to emphasize the need for disrupting internal secondary structures for hybridization to occur. Second, we agree that the discussion on high self-folding energy did not covey the intended message well. By “high self-folding energy” we meant algebraically high (i.e., positive) *?**G* of the reaction. Using this definition, a positive value of *?**G* indicates that the equilibrium of the folding reaction is shifted to the left: that is on average, the sequence tends to exist in an unstructured state:

Unstructured ⇌ Folded

*?**G = −RT* ln*(k*_*folding*_*/k*_*unfolding*_)

We have now replaced the phrase “sequences with high self-folding energy” with “unstructured sequences” throughout the text for simplicity.

*11) −1.6 +/− 1.7; is the latter number the SD*?

In the expression “-1.6 +/− 1.7”, “1.7” is the standard deviation (which shows that the self-folding free energy *?**G*_*37°C*_ of a random 4-base 19-mer has a broad distribution).

*12) The description of the complexity index is not straightforward to understand (what does “by cutting and insertion” mean exactly; cutting of what, where, and insertion of what?)*.

We added two sentences and two references to better explain the Lempel Ziv complexity measure.

*13) Could the authors specify the rate of initiation of T7 RNAP in units of M*^*-1*^
*s*^*-1*^
*rather than just s*^*-1*^?

In this work we measured the rate of promoter escape, which is the rate of promoter-bound T7 RNAP entering productive elongation in the presence of NTPs. This rate is independent of the concentration of RNAP in solution, and, thus, was given in the unit of s^-1^.

*14) Are the regions up and downstream of the hybridization target for the DNA probe also unstructured*?

The regions upstream and downstream of the hybridization target for the probe do not need to be unstructured, but we generally examine the secondary structures of the transcripts to ensure that the flanking sequences do not interfere with the accessibility of the designed hybridization targets. To emphasis this, we inserted one sentence into the second paragraph of the Discussion section.